# The Past, Present, and Future of Clinically Applied Chimeric Antigen Receptor-T-Cell Therapy

**DOI:** 10.3390/ph15020207

**Published:** 2022-02-09

**Authors:** Yuki Fujiwara, Toshiki Kato, Futoshi Hasegawa, Muha Sunahara, Yoshie Tsurumaki

**Affiliations:** 1Cell & Gene Therapy, Oncology, Novartis Pharma K.K., 1-23-1, Toranomon, Minato-ku, Tokyo 105-6333, Japan; yoshie.tsurumaki@novartis.com; 2Oncology Medical Affairs Dept, Novartis Pharma K.K., 1-23-1, Toranomon, Minato-ku, Tokyo 105-6333, Japan; toshiki.kato@novartis.com (T.K.); futoshi.hasegawa@novartis.com (F.H.); muha.sunahara@novartis.com (M.S.)

**Keywords:** chimeric antigen receptor, CAR-T-cell therapy, T-cell health/fitness, multidisciplinary team

## Abstract

Immunotherapy represents the fourth pillar of cancer therapy after surgery, chemotherapy, and radiation. Chimeric antigen receptor (CAR)-T-cell therapy is an artificial immune cell therapy applied in clinical practice and is currently indicated for hematological malignancies, with cluster of differentiation 19 (CD19) as its target molecule. In this review, we discuss the past, present, and future of CAR-T-cell therapy. First, we summarize the various clinical trials that were conducted before the clinical application of CD19-targeted CAR-T-cell therapies began. Second, we discuss the accumulated real-world evidence and the barriers associated with applying clinical trials to clinical practices from the perspective of the quality and technical aspects. After providing an overview of all the moving parts involved in the production of CAR-T-cell products, we discuss the characteristics of immune cells (given that T cells are the raw materials for CAR-T-cell therapy) and elucidate the relationship between lifestyle, including diet and exercise, and immune cells. Finally, we briefly highlight future trends in the development of immune cell therapy. These advancements may help position CAR-T-cell therapy as a standard of care.

## 1. Introduction

Immunotherapy represents the fourth pillar of cancer therapy after surgery, chemotherapy, and radiation. Numerous studies on this treatment modality have been conducted to date; for instance, adoptive immunotherapy, in which a patient’s own cells are processed and then administered to the patient again, has been investigated in clinical trials for several types of cancer using various methods, such as tumor infiltrating lymphocyte (TIL) therapy and dendritic cell therapy [1,2]. Chimeric antigen receptor (CAR)-T-cell therapy, in which a *CAR* gene acting specifically on a target antigen is introduced, has shown efficacy against B-cell malignancies by targeting the cluster of the differentiation 19 (CD19) antigen and has become established as a therapeutic method for cancer [3,4,5,6]. The CAR consists of a single-chain variable fragment as the ectodomain, a short hinge, a transmembrane domain, and an endodomain with signaling domains derived from CD3ζ and costimulatory molecules [7]. After showing success in clinical trials for specific cancer types, the research and clinical applications of CAR-T-cell therapy are advancing worldwide.

In 2017, the FDA approved tisagenlecleucel (Tisa-cel; brand name: Kymriah^®^) and axicabtagene ciloleucel (Axi-cel; brand name: Yescarta^®^), targeting CD19 as CAR-T-cell therapy for commercial use. Subsequently, brexucabtagene autoleucel (Brexu-cel; brand name: Tecartus^®^) was approved in 2020 and lisocabtagene maraleucel (Liso-cel; brand name: Breyanzi^®^) in 2021. In addition, the first B-cell maturation antigen (BCMA) CAR-T-cell therapy, idecabtagene vicleucel (Ide-cel; brand name: Abecma^®^), was approved by the FDA in 2021, and the clinical application of CAR-T-cell therapy is expanding. CAR-T-cell therapy is expected to be indicated for different cancer types, and even greater therapeutic effects are likely to be produced by altering the composition of the target molecule and the *CAR* gene [8,9].

In this review, the clinical study results of CAR-T-cell therapy for hematological malignancies and the real-world data (RWD) already introduced in the market are outlined, and the status of the development and clinical studies of CAR-T-cell therapy to date, the current status and issues of CAR-T-cell therapy at medical institutions, and the future prospects are discussed.

## 2. Overview of CAR-T-Cell Therapy Development and Clinical Trials

Adoptive immunotherapy is a therapy that is expected to exert an antitumor effect by artificially amplifying immune cells outside the body and returning them to the patient. A small number of T cells specific for tumor-associated antigens (TAAs) are present in patients and function as anti-tumor immunity [10]. Therapies have been performed to amplify and return these TAA-specific T cells to the patient; however, the frequency of TAA-specific T cells is very low, and their capacity to proliferate in vivo and their long-term survival after administration are limited, limiting their efficacy. In order to solve these problems, the development of chimeric antigen receptors that artificially combine cancer-antigen-specific antibodies and T cells has been in progress since the middle of the 1980s. In 1993, Eshhar and colleagues developed a CAR within T cells that combines an antibody binding domain to recognize a cancer antigen and a T-cell receptor that activates the cells [11]. This represents the first-generation CAR. In the extracellular domain, it has a single-chain antibody (scFV) consisting of a light chain and a heavy chain, both of which are variable regions for monoclonal antibodies against TAAs. The remainder of the CAR includes a transmembrane domain and a ζ chain of the T-cell receptor in the endodomain within the cell. This first-generation CAR showed good antitumor activity in vitro and in tumor-bearing mouse models but failed to show significant antitumor activity in clinical trials [12]. One of the reasons for this is the lack of a non-specific T-cell activation signal (signal 2) via co-stimulatory molecules. T cells typically require a specific stimulus (signal 1) from the T-cell receptor as well as a nonspecific signal (signal 2) from a costimulatory molecule on the antigen-presenting cells for an immune response [13]. However, since tumor cells often have reduced expression of the co-stimulatory molecules, the T-cell activation may not be sufficient even if the CAR-T cells are able to recognize the target antigen, causing the CAR-T cells to become immune unresponsive. In an effort to solve this problem, a second-generation CAR was developed, in which a second signal of T-cell activation, such as the T-cell costimulatory molecules CD 27, CD 28, 4-1 BB, and OX 40, was directly incorporated into the *CAR* gene [14,15]. As the second-generation CAR combines the CD3ζ chain and a costimulatory molecule in the intracellular signal region, recognition of the target antigen by the scFV results in both the first (CD3ζ) and the second (costimulatory) signals being generated. All the approved CAR-Ts are now considered to be of this second generation, and evidence of therapeutic activity has been demonstrated in clinical studies with second generation CARs using either of the two costimulatory molecules, CD 28 and 4-1 BB. A third generation of CARs that incorporate two costimulatory molecules, such as CD 28 and 4-1 BB or the combination of CD 28 and OX 40, has also been developed.

Numerous cell therapy clinical trials are registered on ClinicalTrials.gov, a public online database and the largest registry of clinical studies conducted in the US and worldwide. According to this database, the first CAR-T-cell therapy clinical trial was conducted in 1997 with a first-generation CAR-T for ovarian epithelial cancer. Since then, the number of trials has continued to increase, with 52 studies performed between 2010 and 2014, 259 studies between 2015 and 2017, and 480 studies between 2018 and 2020. This increase is primarily related to the clinical trials registered in China and the US, which account for 87% of all the trials. The number of CAR-T clinical studies is predicted to increase 1.7-fold per year in the US, 2.12-fold in China, and 1.34-fold in other countries by 2025. Therefore, it is anticipated that CAR-T-cell-based therapeutic platforms will expand in the future [16]. Of the clinical studies conducted, 63% targeted hematological malignancies and 37% targeted solid tumors. In the beginning of 2010, clinical studies targeting leukemia and lymphoma accounted for the majority of studies, but in recent years, studies targeting multiple myeloma (MM) and solid tumors have been increasing, and the indications for CAR-T have been expanding based on the success of CD19-targeted CAR-T-cell therapy. Common targets for hematological malignancies and solid tumors include 20 antigens under development, including CD19, BCMA, MUC1, GPC3, ROR1, and CD20 [17]. The development of CAR-Ts from the second generation onwards, in which sustained effects are produced with a single infusion, the rise of studies in China and the US, and the growth in the number of target antigens due to advances in molecular biology have all contributed to the ongoing expansion of CAR-T-cell therapy.

## 3. Establishment of an Ecosystem for Actual Clinical Use of Regenerative Medicine Products

CAR-T-cell therapy is expected to be widely used in hospitals; however, to provide such therapy, both the manufacturing and the medical sides must establish an ecosystem for collecting raw materials, preparing the product, administering it, and following up with patients. The currently marketed CAR-T-cell therapies are autologous therapies using the patient’s T cells as the raw material, and thus there are many steps involved. This includes (1) transfer from the patient’s general hospital to a hospital handling CAR-T-cell therapy (if necessary); (2) determination of eligibility for CAR-T-cell therapy; (3) apheresis for collection of raw T cells (if eligible); (4) preparing and cryopreserving apheresis products at the hospital (if necessary); (5) product processing to obtain CAR-T-cell therapy products at the manufacturing site; (6) receipt and storage of CAR-T-cell therapy products at ultra-low temperatures at the hospital; (7) lymphodepleting chemotherapy (if necessary); (8) administration of CAR-T-cell therapy products; (9) safety management, including monitoring the characteristic adverse reactions of cytokine release syndrome (CRS) and neurological events; and finally (10), discharge of the subject to his/her general hospital (if necessary) [18]. Some of these procedures are the same as for bone marrow and hematopoietic stem cell transplantation, which are mainly performed for patients with hematological diseases. Although existing operations and systems for bone marrow and hematopoietic stem cell transplantation can be used in hospitals, there is a need to develop new processes for CAR-T-cell therapy [19]. These therapies cannot be handled only by the attending physician or nurses in the department concerned but should be handled by a multidisciplinary team. The hospital systems and multidisciplinary teams required for CAR-T-cell therapy are described below.

CAR-T-cell therapy involves two major phases: (1) the collection, preparation, and storage of raw materials/patient T cells of suitable quality for manufacturing the CAR-T-cell therapy products, especially for post-marketing, under good manufacturing process/good gene, cellular, and tissue-based product manufacturing practice and (2) the selection of suitable patients for receiving CAR-T-cell therapy, followed by appropriate safety management after administration. For the first phase, it is necessary to create an environment in the hospital where these activities can be performed at a level appropriate for the regulations of each country/region and at a level required by the pharmaceutical companies manufacturing CAR-T-cell products. As there are standards and certifications for cell therapy put forth by regulatory bodies, such as the Foundation for the Accreditation of Cellular Therapy (FACT) in the US and FACT-JACIE (The Joint Accreditation Committee ISCT-Europe & EBMT) in the EU, it is necessary to prepare a hospital in terms of equipment, facilities, personnel, documents, and operations to satisfy the requirements for the raw material collection in compliance with these standards [19,20]. For the second phase, it is necessary to at least ensure that there are specialists in the relevant disease on the team and that the team is trained for appropriate safety management. Moreover, cooperation should be secured with departments other than those of the specialists, such as the intensive care unit (ICU) and the neurology department, before the operation [19]. In addition, the involvement of administrative departments, such as legal affairs and finance, are also necessary for the hospital to establish new treatment modalities [21]. When patients receive CAR-T-cell therapy, numerous personnel are involved in the many treatment steps mentioned above [22,23]. Ideally, a multidisciplinary care team for CAR-T-cell therapy should be established in the hospital, and CAR-T-cell therapy should only be initiated after clarifying who is involved in each process and to what extent. However, since it is often difficult to re-assign roles, it will be necessary to determine the roles and operations suitable for CAR-T-cell therapy by referring to bone marrow and hematopoietic stem cell transplantation operations. In addition to disease specialists, multidisciplinary care teams should include senior nurses, coordinators, nurse educators, apheresis nurses, apheresis physicians, apheresis technicians, inpatient nurses, outpatient nurses, cell therapy physicians/technicians, care managers, and social workers. Thus, a multidisciplinary care team for a patient is important to successfully provide CAR-T-cell treatment.

Here, we describe the necessary personnel for each process (1 to 10) mentioned above as an example. (1, 2) In addition to the coordinator, the attending physician, nurse practitioner, and pathologist, if necessary, will be involved in the hospital transfer to determine the eligibility for CAR-T-cell therapy. In particular, the senior nurse, attending physician, and coordinator should communicate closely with doctors from the referring hospital to obtain the necessary information. (3) For apheresis, it is necessary to involve members experienced with the technique, such as attending physicians, apheresis doctors, coordinators, and apheresis nurses. It is also important to cooperate with the physician at the original referring hospital to understand the schedule and status of appropriate treatment before apheresis. (4) For the cell preparation and storage, there is little involvement with the patient directly, but it is essential to cooperate with the cell therapy doctor/technologist who knows the time and date of apheresis and appropriately performs the subsequent work. (5) During the manufacturing of CAR-T-cell therapy products at the manufacturing site, a bridging therapy is often required to control the disease. In such cases, not only disease specialists but also senior nurses and coordinators need to cooperate closely and share the treatment schedule and patient history. (6) Based on the date of receipt of the CAR-T-cell products, the cell therapy doctor/technician must decide on the appropriate date and time for lymphodepleting chemotherapy. (7) For lymphodepleting chemotherapy, cooperation with in-patient nurses and pharmacists, in addition to the attending physician and advanced nurses, is required. (8) When administering the CAR-T-cell therapy products, removing from the freezer, thawing, and administering the frozen products requires the involvement of a cell therapy physician/technician, an advanced nurse, a coordinator, an educational nurse, etc., in addition to the attending physician. (9) Safety management after the administration of CAR-T-cell therapy requires the involvement of primary physicians, advanced nurses, coordinators, educational nurses, inpatient nurses, and social workers, as well as cooperation with ICUs and neurologists, depending on the occurrence of adverse reactions. (10) At the time of discharge from the hospital, the personnel involved in the treatment, such as attending physicians, advanced nurses, nurses, and those who perform the overall follow-up, such as case managers and social workers, should be involved [24]. If the patient visits the hospital on an outpatient basis, it is necessary to continuously involve a senior nurse and coordinator in addition to the attending physician. Hospitals cannot initiate CAR-T-cell therapy operations unless all the procedures are properly arranged. Moreover, the medical care team will vary depending on each hospital and the level of experience with CAR-T-cell therapy. Considering that bone marrow and hematopoietic stem cell transplantation have now become the standard of care for hematological malignancies and that appropriate teams have already been established at hospitals, we hope that such an environment can also be created where patients can comfortably receive CAR-T-cell therapy.

## 4. Real-World Evidence of Clinically Applied CAR-T-Cell Therapy

To date, three anti-CD19 antibody CAR-T-cell products have been approved worldwide: Axi-cel, Tisa-cel, and Liso-cel. Axi-cel was approved in the US/EU in 2017 and in Japan in 2021 for relapsed/refractory (r/r) adult diffuse large B-cell lymphoma (DLBCL) and primary mediastinal B-cell lymphoma (PMBCL). The FDA also approved Axi-cel for r/r follicular lymphoma (FL) in 2021. Tisa-cel was first approved in the US in 2017 for r/r pediatric and young adult B-cell acute lymphoblastic leukemia (B-ALL), in 2018 for the treatment of r/r adult DLBCL, and in the EU in 2018 for the treatment of r/r adult DLBCL and r/r pediatric and young adult B-ALL; approval was granted in Japan in 2019. Liso-cel was approved in the US, EU, and Japan in 2021 for r/r adult DLBCL, high-grade B-cell lymphoma, PMBCL, and FL grade 3B. The pivotal studies on these anti-CD19 CAR-T-cell products have demonstrated their respective efficacy and safety; however, post-marketing RWD are also attracting attention because background information, such as patient demographics and the grading scale for adverse effects, differed among the studies. Table 1 summarizes the studies that have undergone peer review from the time of approval to September 2021; PubMed and Google Scholar were used for the literature search. As Liso-cel has been recently approved, few studies on RWD are available; therefore, we mainly describe the Axi-cel and Tisa-cel cases.

Table 1 lists the number, age, disease, efficacy, and safety profiles of patients treated with each CAR-T product in the real-world (RW) setting. As Axi-cel was the first product to be approved for r/r DLBCL, there are numerous reports on the RWD up to 2021. It is difficult to make a simple, side-by-side comparison because each study was conducted independently of the others. However, compared to the pivotal study ZUMA-1, the Axi-cel RW setting has a slightly higher median age of treatment, and bridging therapy other than corticosteroids is often administered prior to infusion of the CAR-T-cell product. Despite these differences in patient characteristics, the efficacy and safety have generally been shown in the RW setting to be similar to that in the pivotal study. Among the examples of RW reports in Table 1, the following reports are of note:

In a report of eight patients treated with radiation as a bridging therapy, one patient showed complete response (CR), two displayed a partial response (PR), two showed stable disease (SD), and one showed progressive disease (PD); the remaining two patients were not evaluable after 30 days of CAR-T-cell infusion. This study demonstrates that radiotherapy can be safely used as a bridging therapy [26]. In the RW experience, bridging therapy may be needed in terms of disease progression, and the main purpose of the bridging therapy is disease control and/or debulking disease to provide CAR-T therapy. A previous report referring to the role of bridging therapy in CAR-T therapy suggested that bridging therapy should be considered for non-Hodgkin’s lymphoma (NHL) patients who have bulky disease (≥10 cm), >1 extranodal site involved, stage 3–4 disease, bone marrow involvement, and elevated pretreatment lactate dehydrogenase and C-reactive protein [39]. Systemic chemotherapy is usually selected as a bridging therapy, but radiation therapy could be also an optional treatment when patients are eligible for radiation therapy. Some reports have shown the utility of radiation therapy as a bridging therapy, possibly because it is less likely to cause side effects and damage the patients’ lymphocytes than systemic chemotherapy [32,40]. In a report of 283 patients undergoing apheresis, 262 received Axi-cel infusion, and the overall response rate (ORR) was 54% at three months post-CAR-T-cell infusion, while the median event-free survival (EFS) was 9.5 months. The incidence of adverse reactions was 6% for grade ≥3 CRS and 31% for grade ≥3 immune effector cell-associated neurotoxicity syndrome (ICANS). In the same report, 17 patients with secondary central nervous system (CNS) disease who were excluded in the ZUMA-1 study were analyzed separately from the 283 patients; EFS was better in the group without CNS disease than in the group with CNS disease [30]. For the four relapsed DLBCL subjects who underwent allo-hematopoietic cell transplantation (HCT), the ORR and CR were 75% and 50%, respectively, one month after CAR-T-cell infusion, without grade ≥3 CRS and ICANS. Although the sample size was limited, the ZUMA-1 trial excluded allo-HCT patients, and it suggested that the use of CAR-T cells after allo-HCT is generally safe and does not seem to worsen graft-versus-host disease [31]. In a report of 124 patients using radiotherapy and systemic chemotherapy as a bridging therapy, progression-free survival (PFS) and OS at 12 months were 37% and 64% at grade ≥3 CRS and 9% and 40% at grade ≥3 ICANS, respectively [32]. This report showed that radiation as a bridging therapy of CAR-T cells led to good efficacy compared to systemic chemotherapy, despite its retrospective study. A significant difference in PFS and OS between the bridging therapy and the no-bridging therapy cohort can be noted here. A study on Axi-cel treatment of 10 patients with r/r DLBCL, including two with HIV or hepatitis, showed that the CR at three months was 80%, the OS at the data cutoff was 80%, and the grade ≥2 CRS and ICANS were 20% and 30%, respectively, and no severe adverse reactions were observed in patients with HIV or hepatitis [33]. Patients with a history of hepatitis C virus (anti-HCV positive) infection were excluded from the pivotal study. In general, patients who are candidates for CAR-T therapy are pre-treated with multiple doses of chemotherapy and radiation therapy, are immunocompromised, and are at high risk for infections. As the virus may increase after CAR-T-cell administration, these patients should not be administered with CAR-T cells in terms of the proper use of medicine. However, these are important findings in actual clinical practice. In a prospective study of 21 patients, the ORR at 30 days was 67%; the PFS and OS at 12 months were 37% and 49%, respectively; and the grade ≥3 CRS and ICANS were 14% and 19%, respectively [38]. 

The RWD of Tisa-cel have been reported since 2019 (Table 2). As with Axi-cel, while the median age of the RW Tisa-cel treatment is slightly higher compared to that of the pivotal trials, the bridging therapy is used in both the JULIET trial (92% of patients underwent bridging therapy) and in the RW setting. In addition, although it is suggested that there were no major differences in efficacy, the incidence of CRS was generally lower in the RW studies than in the JULIET study [41]. However, it is necessary to consider that the grading scale for adverse effects differed among the studies. In the RW experience, the early referral of patients to a CAR-T treatment facility and/or the early timing of tocilizumab administration may also be contributing factors. Among the examples of RW reports in Table 2, the following reports are of note:

In a report using bendamustine as a lymphodepleting chemotherapy for 28 patients, the ORR and CR at three months were 46% and 38%, respectively; the OS at six months was 71% and grade ≥3 CRS and ICANS were 0% and 4%, respectively [42]. This suggested that the efficacy/safety of CAR-T cells with bendamustine as a lymphodepleting chemotherapy is similar to that with the fludarabine/cyclophosphamide regimen. Tisa-cel treatment of eight cases of NHL with secondary CNS lymphoma who were excluded in the JULIET study showed an ORR of 50%, a CR of 25%, and no grade ≥3 CRS and ICANS events. This suggested that the efficacy/safety of CAR-T cells with bendamustine as a lymphodepleting chemotherapy is similar to that with the fludarabine/cyclophosphamide regimen. Tisa-cel treatment of eight cases of NHL with secondary CNS lymphoma who were excluded in the JULIET study showed an ORR of 50%, a CR of 25%, and no grade ≥3 CRS and ICANS events [43]. This study suggested that Tisa-cel penetrates into the CNS and has anti-tumor activity against CNS lymphoma. For 155 NHL subjects in CIBMTR, the ORR was 62%, the CR was 40%, the PFS/OS at 12 months was 26%/56%, and grade ≥3 CRS/ICANS was 5%/5% [44]. This is one of the largest registry studies of Tisa-cel and shows that the same efficacy as the pivotal study was confirmed in the RW setting. The differences in CRS rates between the RW setting and the pivotal study may be due to differences in grading systems and the timing of tocilizumab administration, as described above.
pharmaceuticals-15-00207-t002_Table 2Table 2Real-world experiences of Tisa-cel.ReferenceN Number of InfusionsMedian Age (Range)Histology (DLBCL/tFL/HGBCL)ORR/CRPFS/OSCRS Any Gr/≥ Gr3ICANS Any Gr/≥ Gr3MEDIAN FURemarksJaglowski S, Blood 2019 Suppl. [45]7065 (19–89)63%/-/31%60%/38% ^a^-/--/4% ^b^-/4% ^b^-CIBMTR registryJakub Svoboda, Blood 2019. [42]2866 (38–81)64%/36%/-46%/38% (3 mo)52% (3 mo)/71% (6 mo)29%/0% ^c^7%/4% ^d^5.5 moBendamustine as LD chemotherapyMatthew J. Frigault, Blood, 2019. [43]850 (17–79)63%/-/24%,  PMBCL 13%50%/25%-88%/0% ^e^13%/0% ^e^-Secondary CNS LymphomaMarcelo C. Pasquini, Blood Adv 2020. [44]15565 (18–89)55%/27%/-62%/40% (BOR)26% (12 mo)/56% (12 mo)45%/5% ^b^18%/5% ^b^11.9 moCIBMTR registryGloria Iacoboni, Cancer Med 2021. [46]75 (Apheresis: 91)60 (52–67)58%/23%/15%60%/32%32% (12 mo)/10.7 mo (median)57%/22% ^b^20%/11% ^b^14.1 mo
S. J. Schuster, NEJM 2019. [41]11156 (22–76)79%/19%/-52%/40%EFS 35% (12 mo)/48% (12 mo)71%/5% ^c^15%/1% 28.6 moPivotal trial (JULIET study)Marcelo C. Pas-quini, Blood Adv 2020. [44]25513.2 (0.41–26.17)B-ALL85%(BOR)EFS 52% (12 mo)/77% (12 mo)55%/16% ^b^27%/9% ^b^13.4 moCIBMTR registryS. L. Maude, NEJM 2018. [47]7511 (3–23)B-ALL81%/61%EFS 57% (12 mo)/77% (12 mo)77%/48% ^c^40%/13%13.1 moPivotal trial (ELIANA study)^a^ 47 patients were evaluated; ^b^ ASTCT consensus grading; ^c^ Penn scale; ^d^ CARTOX; ^e^ Lee scale. B-ALL: B-cell acute lymphoblastic leukemia; LD: lymphodepleting.


Several studies have combined Axi-cel and Tisa-cel analyses (data not shown in the table). In a retrospective study in the US, Axi-cel was administered to 28 patients and Tisa-cel to four patients; the ORR and CR at three months were 59% and 32%, respectively, with grade ≥3 CRS/ICANS of 13%/34% [48]. In a report of 49 patients who received Axi-cel (*n* = 19) and Tisa-cel (*n* = 30), the response rate including SD was 55%, with a median follow-up of 3.4 months [49]. In a report of 163 and 79 patients receiving Axi-cel and Tisa-cel, respectively, the ORR/CR was 59%/44% and grade ≥3 CRS/ICANS was 1%/3% [50]. In a study evaluating 47 Axi-cel- and 22 Tisa-cel-receiving patients, the ORR and CR were 72% and 52%, respectively, with a 12-month PFS/OS of 44%/64% [51]. As each CAR-T-cell therapy reports actual clinical use experience as a class effect, it does not directly compare efficacy or side effects. In another report of Axi-cel (*n* = 45) and Tisa-cel (*n* = 8) therapy, the ORR and CR were 79% and 64%, respectively, with a 12-month PFS/OS of 44%/55%; as for adverse reactions, grade ≥3 CRS and ICANS accounted for 6% and 19%, respectively [52]. In a study of 36 Axi-cel- and 13 Tisa-cel-receiving patients, the CR at 100 days after CAR-T-cell infusion was 51%, followed by a six-month PFS and OS of 48% and 71%, respectively; as for adverse reactions, grade ≥3 CRS and ICANS accounted for 10% and 20%, respectively [53]. In a report on 52 Axi-cel- and 8 Tisa-cel-receiving patients, adverse reactions of grade ≥3 CRS and ICANS were reported in 3% and 20% of patients, respectively [54]. In the median 6.7-month follow-up study in Italy, 18 and 12 patients administered Axi-cel and Tisa-cel, respectively, had ORR/CR rates of 73%/40%. The median PFS was 11.8 months, whereas the median OS was not reached within the follow-up period. As for adverse effects, grade ≥3 CRS/ICANS was 10%/17% [55]. Another study reported that after 49 and 11 patients received Axi-cel and Tisa-cel, respectively, the ORR/CR at three months was 63%/25%, while the median PFS and OS were 3.1 and 12.3 months, respectively [56]. As for adverse effects, grade ≥3 CRS and ICANS were 5% and 11%, respectively. Additionally, there were variable clinical outcomes in the RW setting of each CAR-T product. This is not unambiguously clear but may be due to various factors, such as baseline tumor volume, histology, gene rearrangement, the timing of the CAR-T administration, and the clinical outcome of the bridging therapy. These factors are not identical in each study and clinical outcomes would vary accordingly. In a report by CIBMTR, the best overall response (BOR), 12-month EFS, and OS were 85%, 52%, and 77%, respectively, for 255 patients with ALL; adverse reactions included 16% grade ≥3 CRS and 9% grade ≥3 ICANS, with a median follow-up period of 13.4 months [44]. The results were generally similar between the ELIANA and RW studies for patients with ALL, but the incidence of grade ≥3 CRS tended to be lower in the RW study than in the ELIANA study; nevertheless, the difference in grading scale should be considered [47]. While long-term follow-up data were reported in some clinical studies, the long-term efficacy and safety data of CAR-T cells in the RW are still limited, and analysis of future long-term follow-up data is expected.

## 5. T-Cell Health/Fitness for CAR-T-Cell Therapy

The immune system protects the body from harmful pathogens, such as bacteria, fungi, parasites, and viruses by distinguishing between self and non-self in the body. The concept of immunological surveillance was formulated by Frank M. Burnet in 1970 [57], and Steven A. Rosenberg applied this theory to the clinic as adaptive immunotherapy in 1987 [58]. In 2002, Gavin P. Dunn reviewed cancer immunoediting, which is a mechanism leading to the emergence of immune-resistant variants and highlighted a research direction for assessing why cancer is not eliminated from the body in spite of it being non-self [59]. Cancer heterogeneity is a key challenge of cancer therapy, with cancers being diverse in terms of both genomes and proteins [60]. 

The CAR-T-cell therapies used in clinical practice to date are derived from the autologous peripheral blood mononuclear cells (PBMCs) of patients with cancer, with leukapheresis performed at medical institutions. At the manufacturing site, the T cells are selected, after which the *CAR* gene is introduced using a viral vector. The T cells are engineered to undergo human leukocyte antigen-independent signaling via antigen-specific receptors. After cell culture, the material obtained is released to the medical institution after meeting the release criteria. After the product containing CAR-T cells is infused, it proliferates in the body and results in cancer cell elimination. The CAR-T cells remain in the body, sustaining its effects, such as immunological surveillance of its target antigen. During this process, there are two critical steps that may hinder the CAR-T-cell therapy. The first is determining whether the production process increases PBMCs as the raw material for the CAR-T-cell therapy. It cannot be ruled out that anticancer drug therapy damages T cells. For example, comparing the expression of CD27 and CD28 in the T-cell subsets from healthy subjects, untreated DLBCL patients, and treated DLBCL patients with multiple rounds of chemotherapy, such as with alkylating agents, showed no difference between the healthy subjects and the untreated DLBCL patients, whereas the heavily treated DLBCL patients were double negative for CD27 and CD28 [61]. Both CD27 and CD28 are known markers of senescence, and their expression decreases with age. This suggests that multiple treatments may exhaust T cells, preventing them from receiving proliferation signals during manufacturing. Thus, the material cannot be released as a medicine unless the number of CAR-T cells is increased. The second step is determining whether the infused CAR-T cells can continue to proliferate and survive in the body. For example, a comparison of responders versus non-responders to CAR-T-cell therapy in patients with chronic lymphocytic leukemia (CLL) showed that the CAR-T cells continued to expand and persist after treatment in responders but not in non-responders [62]. For autologous CAR-T-cell therapy, the heterogeneity of the raw materials, including T cells, is a topic for future research. A comparison of individual patient demographics, RNA sequencing, and T-cell phenotypes showed that more T-cell populations expressed CD45RO^−^ CD27^+^ CD8^+^ as a raw material in responders with CCL than in the non-responders. The CD45RO^−^ CD27^+^ CD8^+^ T cells are lymphocyte populations with persistent and long-lasting memory. There was also a difference in the CAR-T cells themselves, and the number of pre-dose CAR-T cells in the CLL responders was limited by the exhausted PD-1^+^ CD8^+^ T-cell population. Differences in the CAR-T cells after treatment were also observed: the CAR-T cells of the responders had many memory T-cell-related genes, such as increased *IL-6*/*STAT3* signatures, and the non-responders had increased gene expression related to effector differentiation, glycolysis, exhaustion of immune cells, and apoptosis. 

Even if the first critical step is overcome, the heterogeneity of the raw materials and CAR-T cells before and after administration influences the therapeutic response. In considering these two potential pitfalls, it is important to obtain an understanding of the proliferation rate of the immune cells, which are the raw materials of CAR-T-cell therapy. In the following sections, we will review the relationship between the immune system, lifestyle (including diet and exercise), and prior therapy before leukapheresis (Figure 1).

### 5.1. Aging and Immune Cells

Lymphocytes, which are components of the immune system, play a crucial role in the immune response against antigens that invade the body. However, this immune function gradually decreases with age. Age-related chronic inflammation, termed “inflammaging”, is deeply involved in age-related diseases, such as lifestyle-related diseases and cancer [63]. 

The thymus gland, a primary lymphoid organ, plays a crucial role in the aging process of immune cells. The thymus gland is several centimeters in size and is located in the chest between the lungs, is larger in children than in adults, and changes from pink to yellow with age. Until J.F.A.P. Miller described the function and role of the thymus gland in 1961, it was considered a cemetery of immune cells [64]. Instead, the T in T cells stands for the thymus because these cells are predominantly produced there. The thymus consists of a cortex and medulla, which play important roles in the recognition of self and non-self. Cortical epithelial cells are responsible for T-cell fate decisions and positive selection by acquiring responsiveness to MHC molecules, subsequently forming a repertoire of cells that can respond to a wide variety of antigens. Medullary epithelial cells contribute to self-tolerance through the negative selection of self-responsive T cells and by producing regulatory T cells. 

Naïve T cells that circulate in the bloodstream through the primary lymphoid organs adhere to the high endothelial venule via adhesion molecules, such as CCR7 and LFA-1 [65], homing to the secondary lymphoid organs and mainly residing in the periarterial lymphatic sheath [66]. Naïve T cells in the lymph node phagocytose antigens from various tissues and repeatedly contact mature antigen-presenting cells, such as dendritic cells and macrophages, to form immune synapses with antigen-presenting cells and non-self peptides that meet the specificity for T-cell receptor (TCR) recognition [67]. Via the immune synapses [68], T cells regulate gene expression through various signaling pathways, including the NF-κB, NF-AT, PI3K/Akt/mTOR, and MAPK pathways, via TCR signals and costimulatory molecules, such as CD28 and 4-1BB (CD137) [69,70,71]. T cells activated by antigens proliferate and differentiate; CD4 T cells mainly differentiate into Th1 or Th2 cells, while CD8 T cells mainly differentiate into killer T cells. Many functional T cells terminate the immune response via apoptosis mediated by the Fas ligand. Some activated T cells survive long term as memory T cells that can rapidly induce an immune response after reuniting with their specific antigen and contributing to antigen clearance. Immune memory is composed of not only memory T cells that can respond to antigens for many years, but also various other mechanisms, such as long-term antigen presentation by antigen-presenting cells and memory B cells [72,73]. 

Age-related thymic involution is thought to result from changes in thymic epithelial cells, which cause a decrease in size, tissue architecture, and the number of thymocytes, leading to decreased naïve T-cell output. Such thymic involution is not necessarily bad, and it is thought that the acquired immunity that maintains memory T cells is still effective against infections; accordingly, the strict selection for producing naïve T cells can be avoided [74]. However, with aging, a chronic, low-grade inflammatory state characterized by a senescence-associated secretory phenotype leads to a decrease in T-cell numbers and an increase in the percentage of T cells with exhaustion markers (CD28^−^ CD57^+^ KLRG1^+^ PD1^+^) [75]. The proliferative capacity of T cells decreases, and the cytotoxic capacity of CD8 T cells also decreases [76]. If the T cells are exhausted due to chronic inflammation caused by chronic infection or cancer, then cell proliferation, cytokine release, loss of memory markers, such as CD44, CD62L, and CXCR3, and failure of the self-repair mechanism mediated by IL-7 and IL-15 will occur [77]. Notably, exhausted T cells do not completely lose their function, leaving them with the capacity to damage cells and release cytokines and chemokines [78]. 

### 5.2. Diet, Exercise, and Immune Cells

A healthy diet is essential for the proper functioning of the immune system. Adipose tissue limits the number and activity of immune cells under nutrient-poor conditions and allows rapid immune cell activity, such as during infection, when nutrients are available. Adipose tissue normally functions as a reservoir of important nutrients during starvation, but excessive fat accumulation causes chronic inflammation and immune exhaustion [79]. This tissue also contributes to the regulation of whole-body energy homeostasis by secreting various hormones, cytokines, and adipokines, but sees quantitative and qualitative changes in response to changes in nutritional status. Obesity caused by an irregular or excessive diet results in excessive fat accumulation, and excess fat accumulation can result in insulin resistance and type 2 diabetes [80]. Hyperglycemic conditions due to diabetes are thought to cause immune response dysfunction. High blood glucose levels adversely affect the ability of PBMCs to produce cytokines. Indeed, anti-CD3-stimulated PBMCs exposed to high glucose concentrations showed impaired DNA synthesis and reduced production of IL-2, IL-6, and IL-10 [81]. The effects of glucose on PBMCs are both exposure-time- and glucose-concentration-dependent. Although hyperglycemic conditions produce few symptoms, 24–72 h of exposure to hyperglycemic conditions at blood glucose levels that cause thirst, polyuria, and malaise may lead to impaired cytokine release. IL-2, originally designated as the T-cell growth factor, is responsible not only for T-cell proliferation and activation but also for increasing B-cell antibody production, monocyte activation, and natural killer cell proliferation and activation [82]. IL-6 is a pleiotropic cytokine that is transiently produced in response to tissue injury and is involved in acute immune responses [83]. IL-10 is an anti-inflammatory cytokine that inhibits the activity and function of T cells and monocytes, thereby contributing to the cessation of the inflammatory response [84]. Other indiscriminate glycation reactions caused by hyperglycemic conditions decrease the ability of T cells to produce IFN-γ and TNF-α, leading to impaired T-cell functions, which include leukocyte recruitment to sites of infection, macrophage phagocytosis, defense against infection, and antitumor effects [85]. 

A close relationship has been reported between the intestinal microbiota and the immune cells [86]. During obesity, the gut microbiota has a role in inducing systemic inflammation [87]. In animal models, inflammation can be modulated via nutrient-induced changes to the gut microbiota composition [88]. For example, it has been suggested that a Western-style diet may alter the composition of the gut microbiota and adversely affect immunity [89]. In humans, it has been reported that changes in the intestinal microbiota due to dietary style can adversely affect immunity [89], but only the type and number of bacteria in feces were examined, with no direct evaluation of the effects on human PBMCs. In 2020, the dynamics between the human gut microbiota and PBMCs were reported [90]. Although the relationship between T-cell subsets in PBMCs and the gut microbiota has not yet been addressed, these findings indicate that it is possible for an irregular diet and overeating to cause unnecessary damage to the raw materials needed for CAR-T-cell therapy.

Habitual physical activity levels decrease with age and have a significant impact on muscle mass and function. Muscle wasting can result from diabetes mellitus, lung disease, heart failure, renal failure, cancer cachexia, etc., and occurs in the whole body as a result of deteriorated nutritional status [91,92]. Loss of muscle leads to increased mortality and a decline in the quality of life [93]. Exercise not only prevents or slows muscle atrophy but also has a positive impact on the immune system [94]. Muscle contraction, muscle remodeling, and exercise training result in the production of secretory factors, including myokines, growth factors, and cytokines, which exert beneficial effects [95]. VO_2_ max, a measure of aerobic exercise capacity, represents the maximum amount of oxygen that the body can consume during exercise; a high VO_2_ max indicates high aerobic capacity. A relationship between VO_2_ max and senescent T cells has been reported [96]; the proportion of senescent T cells increases with age but decreases at a higher VO_2_ max. This suggests that exercise can reduce the adverse effects of aging on the immune system. Although intense exercise increases the proportion of senescent T cells due to damage by reactive oxygen species (ROS) [97], circulating T cells undergo apoptosis to induce progenitor cell mobilization [98]. Abrupt attempts to engage in aerobic or resistance exercise may be difficult for people without established exercise habits. In such cases, it is beneficial to first shorten the time spent sedentary. In general, the total daily energy expenditure can be divided into resting metabolic rate (RMR), thermogenesis, physical activity, and non-exercise activity thermogenesis (NEAT). The RMR accounts for 50–70% of the daily energy expenditure while thermogenesis has an energy expenditure of approximately 10%. Physical activity accounts for 20–40% of energy expenditure, depending on the activity level, and NEAT is expected to consume approximately 350 kcal/day [99]. Patients with cancer may have difficulty performing the necessary levels of physical activity because of weakness or cachexia caused by treatment. To overcome this, exercise intervention may be performed before the treatment is initiated. This is called “prehabilitation”, and it has been reported that when the VO_2_ max is high before hematopoietic stem cell transplantation, the mortality rate and length of hospitalization decreases [100]. Thus, exercise intervention may have a positive impact on the raw materials used for CAR-T-cell therapy.

### 5.3. Prior Therapy before Leukapheresis

T-cell fitness, an important concept in CAR-T-cell therapy because of its direct relationship with clinical outcomes, refers to the capacity of T cells to respond to homeostatic cytokines and resist “death by neglect”. Suboptimal T-cell fitness was shown to be a significant factor related to primary treatment resistance and poor clinical response. Accumulating evidence indicates that T-cell fitness is affected by multiple factors, which can be categorized into intrinsic-related factors, represented by the characteristics of T cells, and extrinsic-related factors, represented by prior chemotherapy and immunotherapy [101]. This section presents a discussion on the impact of each of these factors on T-cell fitness and consequently on CAR-T-cell therapy clinical outcomes.

As described above, prior chemotherapy induces a decrease in early lineage T cells and an increase in terminal effector T cells, resulting in impaired T-cell expansion, which is positively correlated with the number of chemotherapy cycles [102]. Recent evidence has also indicated the impact of chemotherapy on the proliferative function of mature T cells. Cytotoxic chemotherapeutic agents, such as doxorubicin, cytarabine, and cyclophosphamide induce a dysfunctional status in mature T cells, resulting in impaired CAR-T-cell manufacture and affecting the clinical efficacy of ex vivo adoptive cell therapy [101]. These effects are mediated by mitochondrial respiration, glycolytic function, and the mitochondrial integrity of T cells upon exposure to cytotoxic chemotherapeutic agents. Additionally, chemotherapy induces the expression of T-cell senescence markers, which is also positively correlated with the number of chemotherapy cycles [103,104,105]. These observations suggest that T cells undergo both phenotypic and functional alterations after chemotherapy and affect the clinical efficacy of ex vivo adoptive cell therapy.

Bendamustine is another chemotherapeutic agent that affects T cells. Several studies have shown that bendamustine induces grade 3/4 lymphopenia, predominantly involving T cells, while bendamustine + rituximab results in combined T- and B-cell lymphopenia. Consistent with this, bendamustine was correlated with an increased risk of infections [106]. The use of bendamustine is also associated with decreased naïve T cells, but increased Tregs, in patients with NHL and is involved in impairing the cytotoxicity of T-cell function [107,108]. In addition to prior chemotherapy, prior immunotherapy has also been shown to affect the clinical outcomes of CAR-T-cell therapy. For example, prior blinatumomab therapy was correlated with a significantly higher rate of failure to achieve minimal residual disease (MRD)-negative remission and CD19^−^ MRD, as well as relapse in patients with CD19-dim B-lymphoblastic leukemia [109]. Additionally, prior blinatumomab or inotuzumab is associated with suboptimal clinical outcomes of CAR-T-cell therapy when used as a bridging therapy. A recent study conducted in children and adolescent and young adults (AYAs) focused on r/r B-ALL and showed that prior blinatumomab therapy is correlated with a high risk of early resistance or relapse and a shorter OS [110]. In contrast, prior inotuzumab use was not associated with early failure, cumulative incidence of relapse, or EFS but with a decreased OS. These results may be affected by cohort size, duration of follow-up, and potential confounding factors and should thus be interpreted with caution. Nevertheless, these observations indicate that patients with lymphoma have decreased numbers of early lineage T cells in correlation with successive cycles of chemotherapy and that T-cell expansion potential is expected to remarkably decline in heavily pretreated patients. Prior chemotherapy and immunotherapy may thus have an impact on the response rates after CAR-T-cell therapy by affecting T-cell fitness.

Based on the above, general considerations related to the patient’s condition, disease status, and prior therapy are recommended to be considered prior to leukapheresis. For example, when considering Tisa-cel manufacturing, a washout period to ensure sufficient clearance of the drugs used in prior therapy is recommended before leukapheresis. However, the patient’s condition and disease status take precedence over considering washout periods, as indicated by the Tisa-cel apheresis manual. If the patient’s condition and disease status allow, setting a washout period may help avoid the impact of prior therapy on T-cell fitness and thus increase the probability of successful CAR-T manufacturing and the subsequent response rates. In complex cases, such as in highly progressive disease, early apheresis and cryopreservation at an early phase may help avoid this issue and ensure a good quality of collected T cells, as will be discussed in the next section.

### 5.4. Timely Leukapheresis and Cryopreservation

The presence of dysfunctional T cells in the leukapheresis product negatively affects CAR-T-cell fitness [111]. To avoid this issue, early apheresis may help improve the quality of the collected T cells [112]. Cryopreservation can also be employed, as cryopreserved apheresis provides flexibility when scheduling leukapheresis to provide the healthiest T cells depending on the patient’s health conditions [113]. A recent study has shown that growth kinetics and cell viability were similar between fresh and cryopreserved leukapheresis samples [114]. Furthermore, the product quality and clinical outcomes of CAR-T-cell therapy manufactured using fresh or cryopreserved leukapheresis samples were also comparable [115]. A retrospective analysis of clinical trials showed that CAR-T cells manufactured from cryopreserved leukapheresis samples were as clinically effective as CAR-T cells prepared from fresh leukapheresis samples [116]. Therefore, cryopreservation may serve as a safe and viable strategy to ensure the best quality of collected T cells and avoid the negative impact of prior therapy and other factors affecting T-cell fitness.

T-cell intrinsic-related factors include the proportion of CD4:CD8 T cells, T-cell differentiation, the expression of exhaustion markers, and the T-cell metabolic profile [117]. Several reports have suggested that the CD4:CD8 T-cell ratio is not associated with the efficacy (response and survival rates) and safety (grade ≥3 CRS or neurological events (NE)) of CAR-T-cell therapy (Tisa-cel and Axi-cel) for patients with DLBCL [9]. Regarding T-cell differentiation, an important intrinsic factor impacting CAR-T-cell manufacture and clinical outcome, early lineage T-cell levels decreased while terminal effector T-cell levels increased in the peripheral blood as patients with NHL and ALL underwent chemotherapy cycles. Consequently, the potential for T-cell expansion is negatively correlated with the balance of early lineage/terminal effector T cells and the number of chemotherapy cycles [102]. Conversely, a high frequency of stem-like memory T cells is associated with favorable CAR-T-cell expansion and durable responses in DLBCL patients [115]. T-cell exhaustion is also related to the failure of CAR-T-cell manufacturing and poor clinical outcomes. A recent study has shown that patients with high LAG-3^+^ CD3^+^ T cells were non-responders at month six and had a shorter median duration of response, PFS, and OS than did the patients with low levels of LAG-3^+^ CD3^+^ T cells [118]. In another study, the absolute number of CD8^+^ EGFRt^+^ and CD4^+^ EGFRt^+^ T cells was significantly higher in the functional responders, whereas the dysfunctional responders showed significantly higher levels of LAG-3^+^ and TIM-3^+^ T cells, both in the CD8^+^ EGFRt^+^ and CD4^+^ EGFRt^+^ T-cell subsets [119]. Moreover, a higher frequency of CD8^+^ T cells expressing PD-1 and LAG-3 was observed in the dysfunctional responders compared with the functional responders; however, no differences in the frequency of the TIM-3^+^ CD8^+^ T cells were observed. Similarly, a high frequency of CD4^+^ T cells expressing PD-1 but not TIM-3 in the dysfunctional responders was observed. T-cell fitness is also affected by the metabolic status of the tumor microenvironment (TME), such as nutrient deprivation, hypoxia, and toxic metabolites, which consequently impact T-cell metabolism and function, resulting in suppressed cell division and immune functions [120]. Several approaches may efficiently improve the metabolic condition of CAR-T cells to overcome the immunosuppressive effect of cancer metabolism, such as combining CAR-T cells with enzymes to metabolize ROS and nutrient transporters or activating the metabolic reprogramming pathways in CAR-T cells [121].

## 6. Future CAR-T-Cell Therapy

CAR-T-cell therapy is highly effective and has been reported to produce long-term remission in some cases. The ORRs of the JULIET and ZUMA-1 studies were 52% and 82%, respectively, with a 12-month PFS of 25% and 44%, respectively [9,41]. However, some patients do not respond to CAR-T-cell therapy. Relapse after CAR-T-cell therapy usually occurs within half a year of treatment, but cases of late relapse have also been reported [122,123]. The primary mechanisms of acquiring resistance to CAR-T-cell therapy have been reported to include (1) loss of the CD 19 antigen and (2) suppression of CAR-T-cell function by the TME [124,125]. The following sections will discuss these mechanisms as well as the future CAR-T-cell therapy prospects.

### 6.1. Antigen Loss

The target antigen of CAR has been shown to be cleared in patients who relapsed after CAR-T-cell therapy. Specifically, a frameshift mutation causing deletion of the transmembrane domain of the CD19 antigen is observed in ALL cases that relapsed after CD19 CAR-T-cell treatment [126]. Loss of the CD19 antigen due to the deletion of the exon encoding the epitope targeted by FMC63 or the deletion of the transmembrane domain, which is adopted in many CD19-targeted CAR-T-cell therapies, has also been reported to cause CD19-negative relapses [127]. Gardner et al. [128] showed that CD19 CAR-T-cell treatment results in the relapse of CD19-negative acute myeloid leukemia in B-ALL patients with mixed leukemia gene rearrangement and reported lineage switching with CAR-T-cell selection pressure. Strong selective pressure by CAR-T-cell targeting a single antigen can result in antigen loss, and approaches targeting multiple surface antigens simultaneously are under investigation. Pan et al. [129] conducted a clinical study of sequential CD19/CD22 CAR-T-cell therapy in pediatric patients with r/r B-ALL. A total of 20 subjects were enrolled, and all the subjects were infused with CD19-targeted CAR-T cells, periodically monitored post-infusion, and received CD22-targeted CAR-T cells once the CD19-targeted CAR-T cells were undetectable. All the patients achieved MRD-negative CR after CD19-targeted CAR-T-cell infusion and remained in this state prior to CD22-targeted CAR-T-cell infusion. After CD22 CAR-T-cell infusion, 17 patients maintained CR, three experienced disease relapse, two exhibited CD19 loss, and one showed decreased CD22 expression. Zeng et al. [130] evaluated the feasibility of CD22/CD19-targeted CAR-T-cell sequential therapy in 14 patients with aggressive r/r B-cell lymphoma with GI involvement. Seven patients achieved CR, three achieved PR, and three achieved SD. The six subjects who achieved PR or SD experienced disease progression within two to four months after infusion. In two of these patients, the CD19 and CD22 antigens were lost or downregulated. Of the 14 patients, 13 showed CRS, of which only one was grade 3 and the others were ≤ grade 2. To evaluate the efficacy and safety of the sequential administration of CD19/CD22-targeted CAR-T cells, a total of 89 subjects with B-cell malignancies were enrolled. Of the 51 patients with r/r ALL, 48 achieved MRD-negative CR. Of the 24 patients who relapsed, 23 relapsed with CD19^+^ CD22^+^ and one relapsed with CD19^−^ CD22-dim. The median OS was 31 months. Among the 38 patients with NHL who enrolled, the ORR was 72.2%, with a median OS of 18 months. The frequency of CRS and CAR-T-cell-related encephalopathy syndrome (CRES) was 95.5% (85/89) and 13.5% (12/89), respectively [131]. Dai et al. [132] reported that six patients with B-ALL who received a tandem infusion of CD19/CD22-targeted CAR-T cells all achieved MRD-negative CR and experienced grade 1 or 2 CRS; none of the patients experienced neurotoxicity. Three patients relapsed, and one developed CD22-negative relapse with CD19 downregulation. A deletion in exon 2 of *CD19* in a patient with CD19-negative relapse was identified. Schultz et al. [133] treated 17 r/r B-ALL and 22 r/r LBCL patients with bispecific CD19/CD22-targeted CAR-T cells. For patients with B-ALL, 100% achieved CR, and 88% achieved MRD negativity. For patients with LBCL, the ORR was 62% with 29% CR. The relapses were CD19-negative/low in 50% and 29% of patients with B-ALL or LBCL, respectively, but were not associated with CD22-negative/low disease. The CD19/22-targeted CAR-T cells also showed reduced cytokine production when stimulated with CD22 versus CD19. Amrolia et al. [134] developed Auto-3, a bicistronic CAR-T-cell therapy that targets both CD19 and CD22. A total of ten patients were enrolled, but one patient received only CD19-targeted CAR-T cells and two patients were followed for less than 30 days after the end of treatment; the remaining seven patients achieved CR/CR. Three patients subsequently relapsed, one of whom had CD19-negative/CD22 low expression, nine patients had CRS (either grade 1 or 2), one patient experienced grade 1 neurotoxicity, and no patient experienced grade >2 neurotoxicity. A phase II study of the combination of CD19/CD20-targeted CAR-T cells enrolled 25 subjects with r/r DLBCL, of which 21 were successfully treated with CAR-T cells [135]. The ORR was 81% at three months post-infusion, and 11 patients achieved CR. All the treated patients experienced CRS, six cases of which were grade 3/4. Five patients experienced CRES, of which two cases were grade 3/4. Tong et al. [136] evaluated TanCAR7 T cells, which are tandem CD19/CD20-targeted CAR-T cells, in subjects with NHL. Of the 28 evaluable subjects, 20 achieved CR, two achieved PR, four relapsed, and one patient showed antigen loss. CRS occurred in 14 patients, with 10 grade 1/2 cases and four grade 3 cases. Six patients developed grade 1/2 neurotoxicity. The University of Wisconsin evaluated the feasibility of tandem CAR-T cells targeting CD19 and CD20 prepared using the CliniMACS Prodigy system [137]. At ASCO 2019, the results from a clinical trial of 11 adult patients with r/r NHL were presented. Nine subjects achieved an OR, with six CR and three PR. All the CR patients remained in remission at the time of the data cutoff. All the patients with disease progression had a positive CD19 or CD20 biopsy. Six subjects experienced CRS, and three subjects experienced CRES. However, no grade >3 CRS or CRES was observed [138]. Shah et al. [139] evaluated 22 patients with r/r NHL receiving CD19/CD20-targeted tandem CAR-T-cell therapy. No dose-limiting toxicity (DLT) was observed; one patient experienced grade 3/4 CRS, and three patients experienced grade 3/4 neurotoxicity. By day 28, 82% of the patients had achieved a response (CR: 64%, PR: 18%). Notably, CD19 antigen loss was not observed in the relapsed patients. 

### 6.2. TME and CAR-T-Cell Exhaustion

Loss of the CD19 antigen has been reported as a mechanism of acquiring CAR-T-cell resistance in ALL [140]. However, several reports on relapse following the CAR-T-cell infusion of patients with lymphoma have not shown CD19 antigen loss, and the reports are limited in number [9,41,141]. The TME expresses immune checkpoint factors, such as PD-1. T-cell binding to PD-1 activates an inhibitory signaling pathway that decreases T-cell activity, leading to T-cell exhaustion; in addition, CAR-T cells, similar to T cells, express immune checkpoint factors and receive immunosuppressive signals [142]. Analysis of the JULIET study showed that patients with the highest PD-1/PD-L1 interaction score or the highest percentage of LAG3-positive T cells prior to Tisa-cel treatment did not respond to Tisa-cel or relapsed early with no long-term response [41]. In addition, Fraietta et al. [143] reported that out of the 41 patients with CLL who were treated with CD19-targeted CAR-T cells, those that achieved CR had a significantly lower percentage of CD8^+^ PD-1^+^ cells than the non-responders and those that achieved PR. Moreover, CAR-T cells co-expressing PD-1 with LAG3 or TIM-3 showed poor responses, whereas CAR-T cells administered to patients in long-term remission showed a significantly lower frequency than those in relapsed patients. Suppression of CAR-T-cell function via immune checkpoint factors in the TME may lead to CAR-T-cell exhaustion and contribute to treatment failure [144]. The development of immune checkpoint inhibitors against PD-1, PD-L1, and CTLA-4 has revolutionized immuno-oncological approaches and is being investigated in combination with CAR-T-cell therapy. In 2017, Chong et al. [145] administered pembrolizumab to a patient with r/r DLBCL whose disease progressed one month after receiving CD19-targeted CAR-T-cell therapy. Continuing treatment every three weeks for one year resulted in remission, with an increase in the number of CAR-T cells in the patient and a decrease in the expression of PD-1 on the CAR-T cells. Maude et al. [146] administered pembrolizumab to four patients with r/rB-ALL who had an inadequate response to CD19-targeted CAR-T-cell therapy; although the treatment was successful and resulted in long-term detection of CAR-T cells, it was not durable. In 2019, Ardeshna et al. [147] reported the first results of the Alexander trial; subjects with r/r DLBCL were treated with CD19/22-targeted CAR-T cells (Auto-3) and received pembrolizumab 14 days after CAR-T-cell infusion. In seven subjects who received Auto-3 + pembrolizumab, no DLT or investigational product-related death occurred. Although 27% of patients experienced grade 1 CRS, no grade ≥2 CRS was reported. There was only one case of grade 3 neurotoxicity, and the ORR was reported to be 57% (CR 29%). To date, the above clinical studies registered in ClinicalTrials.gov have employed a strategy of combining CAR-T cells and immune checkpoint inhibition for treating hematological malignancies (Table 3). Other approaches, such as inhibiting the immune checkpoint signaling of CAR-T cells by secreting and expressing checkpoint inhibitor molecules or by knocking out *PD-1* have also been attempted.

### 6.3. Allogenic CAR-T Cells

As the existing CAR-T-cell therapies consider autologous cells as living entities, individualized leukapheresis and manufacturing processes are essential. Although this individualized approach has yielded excellent clinical data, there are issues to be solved, such as the high cost of personalized medicine, the risk of manufacturing failure, and the long manufacturing period (~3 weeks) [148]. In addition, there may be cases where it is difficult to secure a line for apheresis in infants, and there may be cases where a sufficient number of lymphocytes cannot be secured for CAR-T-cell production. Moreover, patients eligible for CAR-T-cell therapy have serious life-threatening diseases that may progress prior to completion of the autologous CAR-T-cell product manufacture, and the biological characteristics of the patient’s own source T cells can also be negatively impacted by prior lines of therapy. To overcome these issues associated with autologous CAR-T-cell therapy, which is currently the standard, allogeneic CAR-T-cell therapy has attracted increasing attention. The allogeneic CAR-T-cell manufacturing process produces batches of cryopreserved T cells that provide patients with immediate access to treatment when needed. Thus, homologous CAR-T cells offer several potential advantages, including cost savings from an industrialization and scale-up perspective. Allogeneic T-cell products require the elimination of endogenous TCRs to prevent graft-versus-host disease in the recipient. This requires knocking out the TCR using a TALEN or CRISPR/Cas9 gene editing approach, performed by transfecting the cells with the *CAR* transgene. In the early clinical trials of allogeneic CAR-T-cell therapy for relapsed or refractory B-cell ALL (UCART19), it was reported that the ORR was 88%, and 86% of responders had no detectable MRD (using flow cytometry or quantitative PCR). Although the short-term results were favorable, there is a possibility that the persistence and efficacy of CAR-T cells may decrease due to subsequent rejection; it is thus necessary to carefully consider this during long-term follow-up [149]. As described above, the development of CAR-T-cell therapy targeting CD19 caused a paradigm shift and has been highly successful in the treatment of B-cell malignancies. While not expressed on hematopoietic stem cells, CD19 is consistently expressed during differentiation from pro-B-cells to mature B cells. This pattern of expression is maintained in B-cell malignancies. Therefore, despite the challenge of hypogammaglobulinemia due to the depletion of normal B cells, the success of CD19-targeted CAR-T-cell therapy is based on the presence of a widely expressed and stable target molecule. However, many issues remain to be solved, even for CAR-T-cell therapy targeting CD19, such as antigen loss, the reduction in the antitumor activity of CAR-T cells by the TME, and the difficulty in producing autologous CAR-T cells. Currently, the development of dual-target CAR-T cells and their combination with immune checkpoint inhibitors are being investigated. Understanding and overcoming spatiotemporal tumor heterogeneity is essential for the future development of CAR-T-cell treatment modalities. An additional challenge with CAR-T therapy is expansion into T-cell malignancies. Currently, all commercially available CAR-T therapies target B-cell malignancies. B-cell aplasia can be managed with immunoglobulin replacement therapy. On the other hand, no effective solution exists for T cell aplasia, which can be much more complex and life threatening than B cell aplasia [150]. As a strategy, it may be worth considering the use of CAR-T therapy as a bridging of allogeneic transplantation, but further research is desired.

## 7. Conclusions

Globally approved CAR-T-cell therapies, such as Tisa-cel, Axi-cel, and Liso-cel, prove that cell therapies can provide long-term efficacy for certain patients with adequate safety profiles. Accordingly, many studies have been conducted to investigate these therapies for other types of cancer by changing the target antigen or further increasing the efficacy by combining CAR-T cells with immune checkpoint inhibitors. Although further development of CAR-T-cell therapy is expected, drug administration involves a complex process that includes product manufacturing and a series of transportation steps. There are also other points to consider, such as the selection of suitable patients for CAR-T-cell therapy, which includes prior treatment status and lifestyle habits. In particular, the development of autologous CAR-T-cell therapy cannot always be uniform as the starting materials used are the patient’s own cells, which are prone to variability. The manufacturing process starts with raw materials of highly variable quality, and hence requires complicated processing to manufacture products that meet quality standards. Even when this feat is achieved, each processing step does have scope for improvement [28]. To further institute CAR-T-cell therapy as a standard of care for cancer, the role of the hospital, including the involvement of hospital personnel, remains to be established. The manufacturing methods also require modification, such as the reduction in manufacturing costs and the involvement of persons involved in the manufacturing process, training, streamlining supply chains, etc. It is also necessary to consider ways of enhancing treatment efficacy via lifestyle and diet. 

We are now at a stage where CAR-T-cell therapy has been proven successful and the relevant treatment procedures are being put into operation. It is expected that future research will lead to the development of CAR-T-cell therapy for a wide range of cancers and with improved efficacy. These advancements will help position CAR-T-cell therapy as a standard of care that is comfortably delivered to the appropriately selected patients.

## Figures and Tables

**Figure 1 pharmaceuticals-15-00207-f001:**
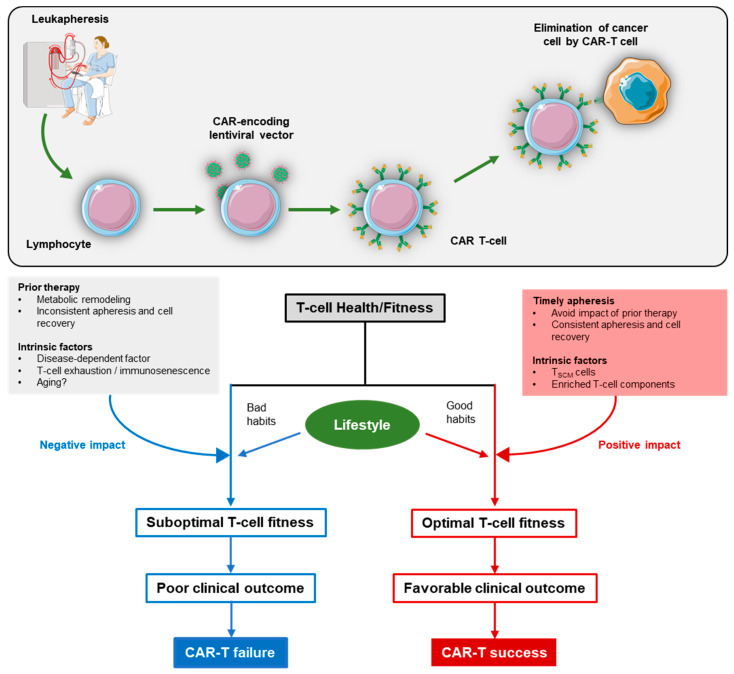
T-cell fitness is impacted positively or negatively by multiple intrinsic and extrinsic factors. A suboptimal T-cell fitness correlates with poor clinical responses, while optimal T-cell fitness is accompanied with increased manufacturing success and good clinical responses.

**Table 1 pharmaceuticals-15-00207-t001:** Real-world experience of Axi-cel.

Reference	N, Number of Infusions	Median Age (Range)	Histology (DLBCL/tFL/HGBCL)	BT Therapy Rates	ORR/CR	PFS/OS	CRS Any Gr/≥ Gr3	ICANS Any Gr/≥ Gr3	MEDIAN FU	Remarks
Loretta J. Nastoupil, Blood 2018, Suppl. [25]	165 (Apheresis: 211)	59 (21–82)	61%/31%/-, PMBCL 8%	56%	59%/49% (Day 100)	-	-/7%	-/31%	-	
Michael D.Jain, Blood, 2018, Suppl. [26]	8	-	DLBCL 1, Double hit 5, tFL 2	100%	50%/17% (Day 30) ^a^	-	86%/14% ^a^	-/43% ^a^	-	RT as Bridging therapy
Caron A. Jacobson, Blood 2018, Suppl. [27]	76	64	DHL/THL 21%	36%	64%/41% (4 mo as med FU, BOR) ^b^	-/84% (4 mo)	96%/17%	76%/38%	-	
Marcelo C.Pasquini, Blood 2019, Suppl. [27,28]	295	61 (19–81)	-/27%/-, DHL 36%	-	70%/52% (BOR)	-	83%/11% ^d^	61%/-	6 mo	CIBMTR registry
Agrima Mian, Blood 2019, Suppl. [29]	27	63 (25–77)	74%/11%/PMBCL 15%	-	-	-/13 mo (median)	-	-	5 mo	
N. Nora Bennani, Blood 2019, Suppl. [30]	262 (Apheresis: 283)	60 (21–80)	67%/-/22%	52%	54% (3 mo)/-	EFS 9.5 mo (median)	91%/6%	67%/31%	10.1 mo	Similar outcome between no CNS and secondary CNS
Tania Jain, Leukemia, 2019. [31]	4	56 (38–66)	DLBCL 3, tFL 1	75%	75%/50% (1 mo)	-	50%/0% ^c^	25%/0% ^c^	112 days	After Allo-SCT
Chelsea C Pinnix, Blood Adv 2020. [32]	124 (Apheresis: 148)	60 (18–85)	77%/16%/-, PMBCL 7%	50%	77%/48%	37% (12 mo)/64% (12 mo) (median)	-/9%	-/40%	11.1 mo	BT cohort/No BT cohort: 50%/50%
Ahmed Abbasi, J Hematol Oncol 2020. [33]	10	66 (55–77)	40%/30%/-	-	-/80% (3 mo)	-/80% (at data cut off)	60%/20% (Gr ≥ 2)	50%/30% (Gr ≥ 2)	-	Two patients with CNS involvement, two patients with HIV and viral hepatitis
Loretta J. Nastoupil, J Clin Oncol 2020. [34]	275 (Apheresis: 298)	60 (21–83)	68%/26%/-, PMBCL 6%	53%	82%/64%(BOR)	47%(12 mo)/68% (12 mo)	91%/7%	69%/31%	12.9 mo	
Caron A. Jacobson, J Clin Oncol 2020. [35]	122	62 (21–79)	43%/14%/27%, PMBCL 7%	45%	70%/50%(BOR)	40% (12 mo)/67% (12 mo)	93%/16%	70%/35%	10.4 mo	
Allison Grana, Clin Lymphoma Myeloma Leuk 2021. [36,37]	37	59 (23–75)	60%/24%/5%, PMBCL 11%	-	49%/35% (6 mo)	5.8 mo (median)/75% (7.5 mo)	97%/16%	73%/43%	11 mo	
Francis A. Ayuk, Blood Adv 2021. [38]	21	58 (24–67)	67%/19%/-, PMBCL 14%	90%	67% (Day30)/-	37%/49% (12 mo)	71%/14%	48%/19%	121 days	Prospective study
Agrima Mian, LEUK& LYMP 2021. [36]	27	63 (25–77)	74%/11%/-, PMBCL 15%	48%	85%/48% (BOR)	10.5 mo/13 mo (median)	-	-	5 mo	
S. S. Neelapu, NEJM 2017. [9]	111	58 (23–76)	76%/16%/-, PMBCL 8%	0%(only corrticosteroids were allowed)	82%/54%	44%/59% (12 mo)	93%/13% ^d^	64%/28%	15.4 mo	Pivotal trial (ZUMA-1 study)

^a^ Seven patients were evaluated; ^b^ Seventy-three patients were evaluable for response; ^c^ ASTCT consensus grading; ^d^ Lee scale. DLBCL: diffuse large B-cell lymphoma; TFL: transformed follicular lymphoma; PMBCL: primary mediastinal B-cell lymphoma; HGBCL: high-grade B-cell lymphoma; DHL: double-hit lymphoma; THL: triple-hit lymphoma; ORR: overall response rate; CR: complete remission; PR: partial response; BOR: best overall response; PFS: progression-free survival; OS: overall survival; EFS: event-free survival; CRS: cytokine release syndrome; Gr: Grade; ICANS: immune effector cell-associated neurotoxicity syndrome; FU: follow-up; Aph: leukapheresis; BT: bridging therapy; RT: radiotherapy; CIBMTR; Center for International Blood and Marrow Transplant Research; CNS: central nervous system; ST: systemic chemotherapy; HIV: human immunodeficiency virus; SCT: stem cell transplantation.

**Table 3 pharmaceuticals-15-00207-t003:** List of clinical studies of CAR-T-cell therapy in combination with immune checkpoint inhibition in patients with hematologic malignancies (6 October 2021).

NCT Number	Disease	Target Antigen	Treatment	Country	Status
NCT02650999	DLBCL, MCL, FL	CD19	Pembrolizumab after CAR-T-cell therapy	USA	Completed
NCT03630159	DLBCL	CD19	Tisa-cel with Pembrolizumab	USA	Completed
NCT02926833	DLBCL	CD19	Axi-cel with Atezolizumab	USA	Active, not recruiting
NCT02706405	NHL	CD19	CAR-T (JCAR014) with Durvalumab	USA	Active, not recruiting
NCT03310619	B-cell malignancies	CD19	CAR T (JCAR017) with Durvalumab	USA	Recruiting
NCT04205409	CLL, FL, DLBCL	CD19	Nivolumab after CAR-T-cell therapy	USA	Recruiting
NCT04850560	B-cell lymphoma	CD19	CAR-T expressing PD-1/CD 28 switch-receptor	China	Recruiting
NCT04381741	DLBCL	CD19	CAR-T expressing IL7 and CCL 19 with PD-1 mAb	China	Recruiting
NCT04134325	HL	CD30	CAR-T-cell therapy with nivolumab/pembrolizumab	US	Recruiting
NCT03298828	ALL, Burkitt lymphoma	CD19	CAR-T-cell therapy with *PD-1* knockout	China	Not yet recruiting
NCT04213469	B-cell lymphoma	CD19	CAR-T-cell therapy with *PD-1* knockout	China	Recruiting
NCT04163302	B-cell lymphoma	CD19	CAR-T secreting mutant PD-1	China	Recruiting
NCT04162119	MM	BCMA	CAR-T secreting mutant PD-1	China	Recruiting
NCT04539444	B-cell lymphoma	CD19/22	CAR-T with Tislelizumab	China	Recruiting
NCT03287817	DLBCL	CD19/22	CAR-T with pembrolizumab	US, UK	Recruiting
NCT03932955	B-cell lymphoma	CD19	CAR-T expressing PD-1/CD 28 switch-receptor	China	Unknown
NCT03540303	B-cell lymphoma	CD19	CAR-T cells carrying cytoplasmic activated PD-1	China	Unknown
NCT03208556	B-cell lymphoma	CD19	CAR-T cells with cell-intrinsic PD1 inhibition	China	Unknown

NCT number: ClinicalTrials.gov identifier; CAR-T-cell: chimeric antigen receptor T cell; CLL: chronic lymphocytic leukemia; PD-1: programmed cell death 1; IL17: interleukin-17; CCL19: chemokine (C-C motif) ligand 19; BCMA: B-cell maturation antigen; DLBCL: diffuse large B-cell lymphoma; B-NHL: B-cell non-Hodgkin’s lymphoma; MCL: mantle cell lymphoma.

## Data Availability

All clinical trial information was obtained from ClinicalTrials.gov (https://clinicaltrials.gov/), PubMed (https://pubmed.ncbi.nlm.nih.gov/) and Google Scholar (https://scholar.google.com/) on 16 September 2021.

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
