# Peer review of "The Past, Present, and Future of Clinically Applied Chimeric Antigen Receptor-T-Cell Therapy"

_pharmaceuticals, 2022, doi:10.3390/ph15020207_

Round 1

Reviewer 1 Report

Dear authors, 

The paper is interesting and covers many aspects of CART-cell therapy. However, I have some comments and suggestions:

  1. The title of the paper is “The Past, Present, and Future…”, but the paper is focused on the present and future only. What I miss is the part about the past of this therapy, how it was developed, the generations of CART-cells, what was the improvement over time, and how the technology looks like now. 
  1. Part 4 – the table is difficult to read, maybe it should be oriented horizontally. I think that the text should be shorter because some information is already presented in the table so there is no point in repeating it. 
  1. Part 5.2 – Are there any studies showing that diet and age influence CART-cell therapy? Are there any markers of T-cell fitness that are taken under consideration when selecting patients for this therapy?  
  1. The future of CART-cell therapy could have a part when the CART-cell strategies for T-cell malignancies are discussed 

Author Response

Thank you very much for your constructive comments. We have updated the entire text based on your comments. Please see the attached file.

  1. The title of the paper is “The Past, Present, and Future…”, but the paper is focused on the present and future only. What I miss is the part about the past of this therapy, how it was developed, the generations of CART-cells, what was the improvement over time, and how the technology looks like now.  
    1. Thank you very much for your valuable suggestions. I have added the correction to the text. 

  1. Part 4 – the table is difficult to read, maybe it should be oriented horizontally. I think that the text should be shorter because some information is already presented in the table so there is no point in repeating it.  
    1. The table has been revised to make it easier to read and the information has been refined. Repetitive explanation of all reported examples in the tables was avoided. Among them, we only discussed those that we wanted to focus on. 

  1. Part 5.2 – Are there any studies showing that diet and age influence CART-cell therapy? Are there any markers of T-cell fitness that are taken under consideration when selecting patients for this therapy?  
    1. There are no reports that answer your questions directly as far as we know. Lifestyle habits such as diet and exercise, aging and the effects of prior therapy before leukapheresis on T cells are being studied. However, the impact on CAR-T cell therapy from these factors needs to be further investigated. In other words, the issue is whether the manufacturing will be successful or not, and even if it is successful, whether the CAR-T cells will be able to function and grow constantly in vivo. We believe that the markers of T-cell fitness suitable for CAR-T cell therapy should be considered. 

  1. The future of CART-cell therapy could have a part when the CART-cell strategies for T-cell malignancies are discussed 
    1. T-cell malignancies were mentioned in the text with references cited. 

Reviewer 2 Report

Pharmaceuticals

The Past, Present, and Future of Clinically Applied Chimeric Antigen Receptor-T-Cell Therapy

The review by Yuki Fujiwara and colleagues describes, with particular care and critical analysis, the various clinical studies that first allowed the use of cell therapy (CAR) -T and then determined the use of this therapy. In this context, the authors report an historical excursus, up to the applications in the clinical field currently in use. The Review is very detailed on the citation of the original studies, it is well structured in paragraphs, each of which reflects the topic with appropriated expertise. The CAR-T topic is of wide interest and the authors provide a clear explanation of the possible variables that may have influenced and still influence the outcome of immunotherapy, thus addressing the related issues, from the removal of cells from the patient to the process. to obtain CAR-T, to infusion and patient management.

The Review is, therefore, well written, well-structured and discussed, with an excellent balance between descriptive text and practical illustrations: it is certainly worthy of publication and, personally, I find it of considerable interest.

  The only minor concern is the lack of an explanation, even if only speculative, with respect to the following questions:

Q1. What explanation or reasons can be considered to explain the reason for the internal variability of efficacy on the effects obtained from the various clinical studies reported? A comment, even if you just report a speculative hypothesis, could add relevance to the context of applying the therapy.

Q2. What role could have played in this sense the different stratification of patients or the fitness of T lymphocytes, which the authors describe very accurately in paragraph 5 (T lymphocyte health / suitability for T-CAR lymphocyte therapy)?

Author Response

Thank you very much for your constructive comments. I have updated the entire text based on your comments. Please see the attached file.

Q1. What explanation or reasons can be considered to explain the reason for the internal variability of efficacy on the effects obtained from the various clinical studies reported? A comment, even if you just report a speculative hypothesis, could add relevance to the context of applying the therapy. 

  1. In addition to the underlying disease that DLBCL patients, multiple cycles of anticancer therapy can exhaust T cells. The aaIPI (age-adjusted international prognostic index) score and PFS are related. At least in the current stage for clinical applied CAR-T cell therapy needs to take into account the patient's T cell status. 

Q2. What role could have played in this sense the different stratification of patients or the fitness of T lymphocytes, which the authors describe very accurately in paragraph 5 (T lymphocyte health / suitability for T-CAR lymphocyte therapy)? 

  1. To describe what role have the different stratification and the fitness of T lymphocytes played in this sense, we added a paragraph that describes general considerations related to the patient’s condition, disease status, and prior therapy are recommended to be considered before leukapheresis. The reviewer’s comment helped us to clarify this role and emphasis the importance of early apheresis and cryopreservation. 

Round 2

Reviewer 1 Report

I accept the authors' responses and have no further comments.